# Strain-engineered inverse charge-funnelling in layered semiconductors

Adolfo De Sanctis [1], Iddo Amit [1], Steven P. Hepplestone [1], Monica F. Craciun[1] & Saverio Russo[1]

The control of charges in a circuit due to an external electric field is ubiquitous to the exchange, storage and manipulation of information in a wide range of applications. Conversely, the ability to grow clean interfaces between materials has been a stepping stone for engineering built-in electric fields largely exploited in modern photovoltaics and opto-electronics. The emergence of atomically thin semiconductors is now enabling new ways to attain electric fields and unveil novel charge transport mechanisms. Here, we report the first direct electrical observation of the inverse charge-funnel effect enabled by deterministic and spatially resolved strain-induced electric fields in a thin sheet of $HfS_2$. We demonstrate that charges driven by these spatially varying electric fields in the channel of a phototransistor lead to a 350% enhancement in the responsivity. These findings could enable the informed design of highly efficient photovoltaic cells.

---

[1] Centre for Graphene Science, College of Engineering, Mathematics and Physical Sciences, University of Exeter, Exeter EX4 4QF, UK. Correspondence and requests for materials should be addressed to A.D.S. (email: a.de-sanctis@exeter.ac.uk) or to S.R. (email: s.russo@exeter.ac.uk)

Manipulating the motion of charge carriers by means of an electric field has been a stepping stone in a wide range of sectors. From electronic circuits to synapses in neural cells[1], the electric field control over the dynamics of charges underpins a vast range of computing, storage, sensing, communication and energy harvesting tasks. For example, built-in electric fields generated at the interfaces between materials in vertical structures govern the extraction of photo-generated carriers in several photovoltaic and opto-electronic applications[2]. Presently, the emergence of atomically thin materials[3] and the development of new ways to tailor their electrical and optical properties, for example by local modification of their composition[4,5] or structure[6], holds the promise to explore new implementations of electric fields and unveil novel mechanisms of charge transport which can boost the efficiency of opto-electronic devices.

The application of strain is one way to engineer electric fields in semiconducting materials through a varying energy gap. However, common bulk semiconductors can only sustain strain of the order of ~0.1–0.4% without rupture[7], a value that limits the range of physical phenomena and applications that can be accessed. On the contrary, layered semiconductors, such as graphene[8] and transition metal dichalcogenides (TMDs)[9], are theoretically predicted to be able to sustain record high levels of strain >25%[10,11] expected to lead to an unprecedented tunability of their energy gap by more than 1 eV[12]. One tantalising charge transport phenomenon that could be accessible owing to large values of strain is the funnelling of photoexcited charges away from the excitation region and towards areas where they can be efficiently extracted[13–15]. Such effect is heralded as a gateway for a new generation of photovoltaic devices with efficiencies that could approach the thermodynamic limit[2,13].

In general, strain-induced gradients of energy gaps create a force on (neutral) excitons that pushes them towards the regions with the smallest energy gap. In direct gap semiconductors, this area corresponds to that of maximum tension. Hence, the strain pattern generated by simply poking a sheet of direct gap TMD would normally funnel the charges towards the apex of the wrinkle[13,14,16,17]. Consequently, the extraction of the charges for energy harvesting or sensing poses considerable technological challenges and for this reason the funnelling effect has not yet been observed experimentally in electrical transport. On the other hand, the opposite behaviour is theoretically expected in some indirect gap semiconductors (e.g. $HfS_2$ and $HfSe_2$) and in black-phosphorus, where the energy gap increases in the regions of tension[12]. This would allow the exploitation of the so-called inverse charge funnelling[18] whereby a strain pattern generated by poking a sheet of these materials would push the charges away from the apex of the wrinkle, making them readily available for energy harvesting or computing purposes to an external circuit.

In this work, we demonstrate the electrical detection of the inverse charge funnel effect using a photo-assisted oxidative method to attain deterministic and spatially resolved electric fields in ultra-thin $HfS_2$. A 350% increase in the responsivity of strained devices compared to pristine structures demonstrates the efficient extraction of photogenerated carriers away from the excitation region. The bias dependence of the photocurrent shows that the measured signal is due to charge funnelling, enabled by the strain-engineered gradient of energy gap in the channel. Our complementary study of a wide range of experimental techniques (i.e. spatially resolved absorption and Raman spectroscopy, elemental analysis and atomic force microscopy (AFM)), together with analytical theoretical modelling and density functional theory (DFT) calculations, confirm that band tailoring by strain in TMDs is a gateway for the observation of novel microscopic charge transport phenomena.

## Results

**Photo-oxidation-induced strain in $HfS_2$.** In traditional semiconductors such as Si and Ge, strain is typically introduced at the growth stage by dislocations or elemental composition[19]. These techniques do not easily allow the creation of complex planar strain patterns, forbidding the development of ultra-thin charge-funnel devices. These limitations can be overcome by using atomically thin semiconductors, such as $HfS_2$. In this case, specific strain patterns can potentially be engineered in the plane of the TMDC by exploiting the lattice mismatch between the semiconductor and its in situ grown oxide, see Fig. 1a. Ab initio DFT calculations suggest that the $[11\bar{1}]$ cleavage plane of monoclinic $HfO_2$ has a spatial arrangement of Hf atoms commensurate to that of the basal plane of $HfS_2$, with an Hf−Hf distance of 3.426 Å. Since the Hf−Hf distance in $HfS_2$ is 3.625 Å, a transition between these two structures is likely to introduce an average 2.7% compressive strain in the semiconductor at the interface with its oxide (see Supplementary Fig. 6). Hence, anchoring the TMD at the edges, for example by depositing electrical contacts[20], would allow the same amount of strain to be induced away from the oxidised area in the opposite direction (green arrows in Fig. 1a). Such tensile strain pattern results in the spatial modulation of the bandgap of $HfS_2$ and therefore the creation of spatially varying electric fields, which are the key ingredients to observe the inverse charge funnel effect. The magnitude of these electric fields can be determined from the change in the energy gap with strain. This has been calculated using DFT and the results, shown in Fig. 1b, c, predict an increasing (decreasing) value of the direct gap $(\Gamma \to \Gamma)$ with compressive (tensile) strain while the indirect gap $(\Gamma \to M)$ behaves the opposite.

To engineer strain-induced electric fields through lattice mismatch we employ a spatially resolved photo-oxidation technique. Upon exposure to a focussed laser $(\lambda = 375$ nm, $P = 1$ MW cm$^{-2}$) we find that $HfS_2$ is readily oxidised, becoming invisible to the naked eyes (see Fig. 2a). Surprisingly, topographic AFM measurements show no ablation of the material in the laser-exposed area while the tapping phase image clearly reveals a change in its viscoelastic properties (see also Supplementary Notes 2–4 and Supplementary Figs. 2 and 3). The energy dispersive X-ray microanalysis (EDXMA) shows the absence of the S peaks (K lines) and the appearance of an O peak (K$_\alpha$ line) in the laser-irradiated areas. This is in stark contrast to the spectrum of the pristine $HfS_2$ where the expected S peaks are clearly measured and no O peak is resolved (Fig. 1b). No change is observed in the Hf and substrate peaks. Quantitative analysis shows that, upon laser irradiation, the weight ratio of S decreases from ~20 to ~1% of the total, while the O content increases from ~1 to ~20%, indicating the formation of $HfO_2$. Furthermore, the oxidised area is compatible with the diffraction-limited spot size of our laser system (see Supplementary Fig. 1d,e), indicating that a photon-assisted oxidation process, as opposed to a thermally driven one, is taking place.

The photo-oxidation of 2D semiconductors has recently been shown to depend on the rate of charge-transfer between the surface of the material and the aqueous oxygen present in air[21] via the oxygen−water redox couple $2H_2O \rightleftharpoons O_2$ (aq) $+ 4e^- + 4H^+$, where the $O_2$ binds to a metal site[22]. Adopting the same model and supported by the EDXMA data, we summarise the oxidation reaction of $HfS_2$ as $HfS_2(s) + 3O_2(aq) + h\nu \to HfO_2(s) + 2SO_2(g)$. Here, the absorption of a photon of energy $h\nu$ produces an optical excitation in the $HfS_2$, leaving it in an excited state $(HfS_2 + h\nu \to HfS_2^*)$ which provides free carriers that are transferred to the oxygen on the surface, producing an oxygen radical ion $O_2^{\cdot-}$ (aq) which reacts with the $HfS_2$ (see Supplementary Note 1 for details). The feasibility of the proposed reaction is

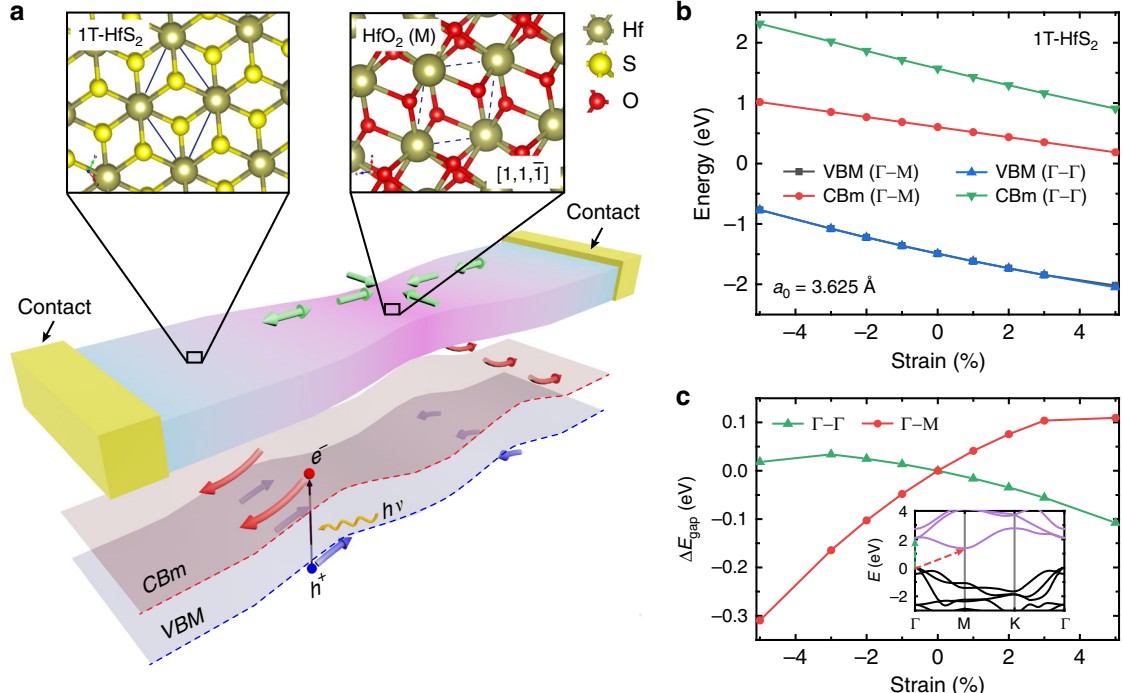

**Fig. 1** Inverse charge funnelling in strained HfS₂. **a** Schematic diagram of the proposed device (see Fig. 4 for the actual implementation). Compressive strain is induced in the centre of a semiconducting HfS₂ channel by controlled photo-oxidation. The compression induces tensile strain away from the HfS₂/HfO₂ interface, resulting in the spatial modulation of the bandgap. **b** Ab initio calculations of the valence band maximum (VBM) and conduction band minimum (CBm) of 1T-HfS₂ as a function of strain in the $\Gamma \rightarrow \Gamma$ (direct gap) and $\Gamma \rightarrow M$ (indirect gap) directions. **c** Change in bandgap as a function of strain in the two directions, with respect to the unstrained bandgap (relaxed lattice constant $a_0 = 3.625$ Å). Inset: calculated band structure of 1T-HfS₂

confirmed by DFT calculations, which show an energy cost of $-11.58$ eV per HfS₂ molecule (see Supplementary Note 6). Indeed, a detailed study of the oxidation rate and its dependence on the laser flux and the initial amount of pristine material confirms that the process is described by the Mercus-Gerischer theory[21] as expected for photo-oxidation (see Supplementary Eq. (4) and Supplementary Fig. 1b,c).

Theoretically, we expect that a 3% compressive strain should induce a change by as much as 30 meV in the bandgap of HfS₂ (Fig. 1c). Indeed, a measurement of the absorption coefficient $\alpha$, in the region close to the laser-written oxide, confirms an energy gap difference of 30 meV (see Fig. 2c, Supplementary Note 5 and Supplementary Figs. 4 and 5). The absorption coefficient measured in the centre of the oxidised area is close to zero, showing that the direct absorption edge lies above 2.9 eV, as expected for HfO₂. Raman spectroscopy allows us to map the strain profile induced in a clamped device, as shown in Fig. 2d inset. First-principle studies have shown that the peak corresponding to the Raman-active A₁g phonon mode of HfS₂ downshifts (upshifts) with the application of tensile (compressive) strain[23]. In Fig. 2d we plot the frequency of such mode as a function of position along the length of the device. The experimental frequency of the A₁g mode of pristine multi-layer HfS₂ is found to be $336.1 \pm 0.01$ cm⁻¹ from the measurements reported in Supplementary Fig. 1c,d, in good agreement with the literature[24]. The deviation of the measured peak from this value demonstrates the presence of compressive and tensile strain along the device (green arrows in Fig. 2d), compatible with the model proposed in Fig. 1a (the data have been calibrated using the position of two fixed peaks to compensate for instrumental shifts, as explained in the Methods section).

**Photoresponse of strain-engineered HfS₂ FETs**. Demonstrating the creation of strain gradients using a spatially resolved photo-oxidation process allows the realisation of novel planar heterointerfaces and energy band tailoring. Hence, we employed scanning photocurrent microscopy (SPCM) mapping[25,26] to study the photoresponse of a strain-engineered HfS₂ photo-detector in a field-effect transistor (FET) configuration in search of the inverse charge funnelling effect. Figure 3a shows the SPCM maps before and after photo-oxidation of a single spot in the channel of the FET. We observe an enhancement of the photo-response close to the laser-oxidised area, where the responsivity increases by 350% at low powers and by 200% at the saturation power (120 W cm⁻²), as detailed in Fig. 3b, see also Supplementary Note 9 and Supplementary Fig. 9 for extended data). In order to correlate this observation with the electrical detection of charge funnelling, we perform SPCM in the device presented in Fig. 2d. Figure 4a schematically depicts the band alignment in such device for an applied bias $V_{sd} = 0$ V, where the changes in the valence band maximum (VBM) and conduction band minimum (CBm) with strain are taken from Fig. 1b for indirect transitions (based on the data from Raman and absorption spectroscopy). Excited electron−hole pairs, in the proximity of the strained area, are funnelled towards the electrodes by the built-in energy gradient, giving an enhanced photoresponse[13,18]. For $V_{sd} = 0$ V, both sides of the strained junction will give equal contribution. Indeed, we were not able to measure any photoresponse in absence of a bias, see also Supplementary Fig. 9. The application of a source-drain bias larger than the difference between the conduction band energy at the maximum strain point and its value in the unstrained region ($V_0$) is expected to exhibit a larger photoresponse in regions with maximal energy gradients (see also Supplementary Fig. 7). This is indeed observed

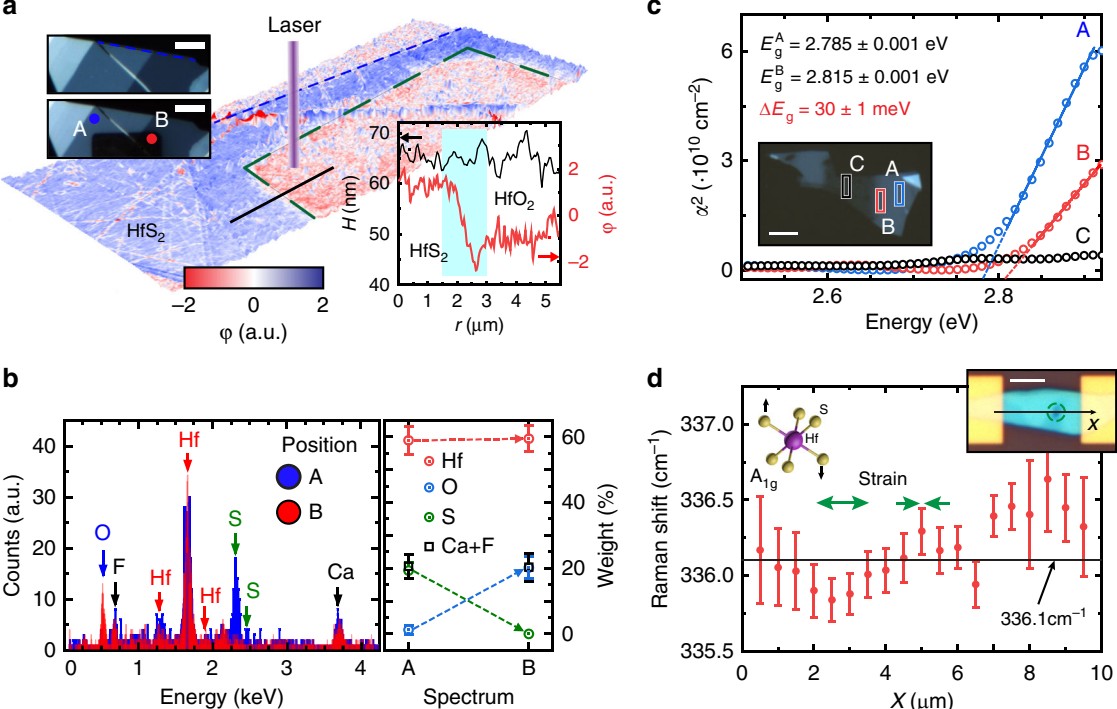

**Fig. 2** Photo-oxidation and strain engineering in HfS$_2$. **a** AFM topography with phase contrast $\varphi$ signal superimposed of a representative flake after laser exposure (green-dashed line). Top-left inset: optical micrograph of the flake before (top) and after (bottom) laser-assisted oxidation. Bottom-right inset: height (H) and phase signal along a 5 μm line-cut (black line). Scale bars are 5 μm. **b** EDXMA spectra acquired in the regions A and B in panel **a** and quantitative analysis of the chemical elements (right). **c** Square of the absorption coefficient ($\alpha^2$) of: (A) HfS$_2$ away and (B) close to the oxidised area and (C) HfO$_2$. Extrapolated direct bandgap: $E_g^A = 2.785 \pm 0.001$ eV and $E_g^B = 2.815 \pm 0.001$ eV, $\Delta E_g = 30 \pm 1$ meV. Inset: optical micrograph of the flake where the colour boxes represent the sampling areas (1 × 3 μm) in which the absorption spectra where acquired. Spectrum (B) is centred at 1 μm from the edge of the oxide area. Scale bar is 4 μm. **d** Frequency of the A$_{1g}$ mode of HfS$_2$ as a function of position along the photo-engineered device shown in the inset (green circle indicates the photo-oxidised area). The horizontal solid line marks the average frequency of an as-fabricated flake at 336.1 cm$^{-1}$, tensile (compressive) strain is marked by a down- (up-) shift of this mode (green arrows). Error bars represent the uncertainty of the Lorentzian fit of the spectra. Scale bar is 3 μm

in the SPCM map shown in Fig. 4d. Furthermore, by reversing the bias it is possible to mirror the profile of the built-in electric field and consequently reflect the position of the maximum photoresponse, see Fig. 4c, e.

**Inverse charge funnelling effect**. To fully capture the role of the inverse charge funnelling effect on the measured photoresponse, we develop a one-dimensional analytical model. For simplicity we assume that the strain gradient induces a built-in potential which decays linearly with the distance from the strain junction, creating a local built-in electric field $E_0$ (see Supplementary Eq. (11)). By solving the charge continuity equation (see Supplementary Note 10), assuming the rate of carrier generation to be a delta function at the illumination point $x_0$, we find that the charge density as a function of position $x$ is given by:

$$\Delta n = \Delta n_0 e^{-\frac{1}{2}\left(\frac{q}{k_b T}(E_{sd} \pm E_0) + \sqrt{\left(\frac{q}{k_b T}(E_{sd} \pm E_0)\right)^2 + \frac{4}{\tau D}}\right)|x - x_0|}, \quad (1)$$

where $\Delta n_0$ is the excited carrier density at the injection point, $T$ is the temperature, $k_b$ is the Boltzmann constant, $q$ is the electron charge, $\tau$ is the carriers lifetime, $D$ is the diffusion coefficient and $E_{sd}$ is the electric field due to the applied bias. The plus (minus) sign applies to the left (right) strain-engineered region, respectively and $E_0 = 0$ outside those regions. Calculating the current generated by this charge density distribution (see Supplementary

Eq. (10) and Supplementary Fig. 10) by scanning the laser along the channel, we can reproduce the experimentally measured SPCM (see Fig. 4f). Although the strain gradient, and thus the built-in field, should be treated as $\propto 1/x$, our simple assumptions allow the derivation of an analytical result which is still able to reproduce well the experimental data with $\tau$ as the only free fitting parameter. In our case we find a value of $\tau \simeq 10^{-10}$ s outside the strain region, which is typical of multi-layer semiconducting TMDs[27]. In the strain region we find $\tau \simeq 10^{-6}$ s, which translates in a carrier diffusion length of $L = \sqrt{\tau D} \simeq 8$ μm (assuming a mobility of 2.4 cm$^2$ V$^{-1}$ s$^{-1}$)[28]. The observation of a diffusion length that exceeds the extension of the strained region (2.5 μm) is, indeed, a signature of efficient separation and extraction of charges, compatible with the charge-funnel effect[14]. Future studies of the effects of bandgap engineering on the carriers recombination lifetimes could elucidate the physical mechanisms behind this improvement and may shed light on the role of hot-carriers in such strained devices for photovoltaic applications. In particular, charge funnelling could allow carriers excited above the bandgap to be extracted before their excess kinetic energy is lost through cooling, enabling solar cells relying on this phenomena to overcome the Shockley−Queisser limit and bring their efficiency above 60%[29]. Furthermore, the spatial modulation of the semiconductor bandgap could be used to create an effective tandem solar cell able to absorb a much larger portion of the solar spectrum compared to a single bandgap device[2] (see also discussion in Supplementary Discussion).

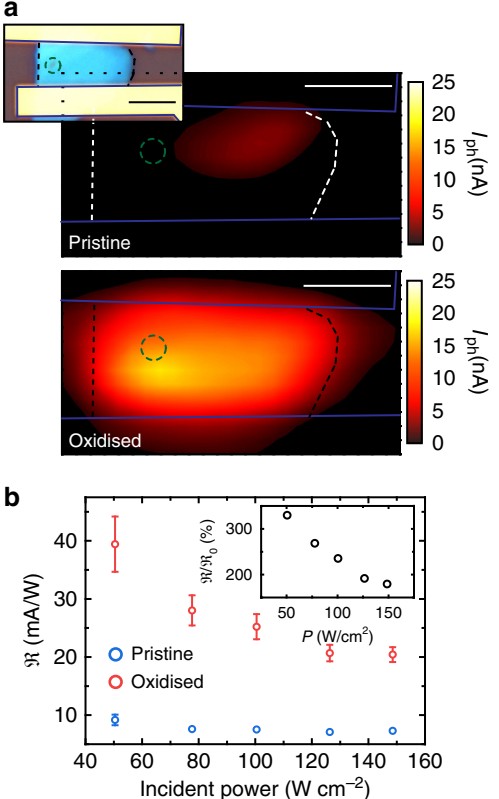

**Fig. 3** Photo-response of $HfS_2/HfO_2$ engineered device. **a** SPCM map of the device before (top) and after (bottom) laser-assisted oxidation. $V_{sd} = -5$ V, $V_{bg} = 50$ V, $\lambda = 473$ nm, $P = 150$ W cm$^{-2}$ and 0.5 μm step size. Inset: optical micrograph of the device, after laser-assisted oxidation of a single spot (green dashed circle). Scale bars are 3 μm. **b** Responsivity before (blue, $\Re_0$) and after (red, $\Re$) laser-assisted oxidation as a function of incident optical power. Inset: ratio $\Re/\Re_0$

## Discussion

In summary, in this work we report the first experimental observation of the inverse charge funnelling effect, that is a novel microscopic charge transport mechanism enabled by strain-induced electric fields. By developing a unique technique of photo-oxidation, we are able to engineer deterministic and spatially resolved strain patterns in ultra-thin films of $HfS_2$ which in return generate built-in electric fields. Such strain gradient is responsible for the enhancement of the responsivity of a phototransistor of up to 350%, which was attributed to the inverse charge funnel effect. A simple analytical model was derived to simulate the SPCM experiments, which demonstrated the charge funnelling effect and allowed the determination of a long carrier recombination lifetime of $10^{-6}$ s in the strain-engineered region of the device. These results open the route towards the exploitation of strain-engineered devices for high-efficiency energy harvesting and sensing applications, with the potential to overcome the intrinsic limitations of current solar cells by exploiting both hot-carriers extraction and lossless transport, to achieve efficiencies approaching the thermodynamic limit in photovoltaic devices[2,29]. The use of atomically thin materials could open the door to the incorporation of such devices in emerging wearable electronics technologies[30] and smart buildings[31], creating a new paradigm in energy harvesting.

## Methods

**Sample preparation.** Thin flakes of $HfS_2$ were obtained by micro-mechanical exfoliation[32] from commercial bulk crystal (HQ Graphene) on different substrates.

The oxygen-free $CaF_2$ substrate was used in the EDXMA allowing one to probe purely the oxygen peak of the oxidised $HfS_2$. This same substrate was used for Raman spectroscopy since it only has a well-defined Raman peak at 322 cm$^{-1}$. Quartz (525 μm thick) substrate was used to perform the absorption coefficient measurements, due to its constant refractive index across the scanned energy range. Substrates of heavily doped Si capped with 285 nm of thermally grown $SiO_2$ were used to fabricate phototransistors using standard electron-beam lithography, deposition of Ti/Au (5/50 nm) for the contacts followed by lift-off in Acetone. For EDXMA analysis the sample was coated with 5 nm of Au and grounded to avoid charging of the $CaF_2$ substrate. The choice of substrate did not affect the laser-induced oxidation in terms of morphology, exposure time or incident power.

**Atomic force microscopy and energy-dispersive X-ray microanalysis.** Atomic force microscope topography and phase image were acquired with a Bruker Innova AFM system, operating in the tapping (or dynamic) mode to avoid damage to the sample while maintaining a high spatial resolution. The measurements were done using a highly doped silicon tip acquired from Nanosensors with a nominal resonance frequency of 330 kHz, and a sharp radius of curvature (<10 nm). Energy dispersive X-ray microanalysis was performed using a Hitachi S-3200N scanning electron microscope equipped with an Oxford Instruments EDS Model 7021 (detection area 10 mm$^2$ and resolution at 5.9 keV of 138 eV). The accelerating voltage was 10 kV and the total counts were fixed at $10^4$ for each spectrum. The observed Hf/S weight ratio (~3) in the pristine area confirmed the stoichiometry of $HfS_2$, as also reported in the literature[33].

**Raman and optical spectroscopy.** Raman spectroscopy of ultrathin $HfS_2$ requires great care in order to avoid the photo-oxidation of the material: low laser intensity, long acquisition time and low background noise are required. For this reason the Raman spectra presented in this work were acquired with a custom-built micro-Raman spectrometer[25]. For very thin flakes we used an incident power density of 5 kW cm$^{-2}$ with an acquisition time of 20 s, with no visible changes in the sample. The same set-up was also used to acquire the transmission and reflection spectra, in order to measure the absorption coefficient, with an incandescent bulb and a white LED as light sources. All light sources were thermally controlled to ensure no thermal drift during measurements.

In order to determine with great accuracy the Raman shift of the $A_{1g}$ mode of $HfS_2$, we calibrated the acquired spectra relying on the presence of two fixed peaks which were acquired in the same spectrum: the silicon peak (from the substrate) and the spurious laser-line peak (L) which appear at 520 and 316 cm$^{-1}$, respectively (see Supplementary Figs. 1b and 2h). Since these two peaks do not belong to the $HfS_2$ they will not shift with the strain applied to the semiconductor and can be used to correct for instrumental shifts of the frequency. The average frequency of the unstrained $A_{1g}$ mode is $336.1 \pm 0.01$ cm$^{-1}$ after averaging 115 spectra (see Supplementary Fig. 1c).

**Determination of the absorption coefficient.** The absorption coefficient $\alpha(\lambda)$ is defined as the fraction of the power absorbed per unit length in the medium, and it is a strong function of the incident wavelength $\lambda$. We used the formulation by Swanepoel[34] to calculate the absorption coefficient of $HfS_2$ and $HfO_2$[25]. In order to account for the interface between the $HfS_2$ and the substrate, we used the measured reflectance of a thick $HfS_2$, so that we can ignore multiple reflections from the substrate, to compute the refractive index $n$ of $HfS_2$. We found that $n \sim 2.5$ across the measured range and thus, $R_2 \sim 5.6\%$ (reflectance at air/medium interface). Since $R_3 = 5.0\%$ (air/quartz interface), we assumed $R_2 = R_3$ in Eq. (A3) in ref. [34]. The same result can be obtained by computing $n$ from the measured transmittance curve, using Eq. (20) in ref. [34]. The bandgap of a semiconductor is related to the absorption coefficient by: $\alpha \propto (h\nu - E_g)^{1/2}$ for direct allowed transitions[35]; therefore, measurement of $\alpha$ close to the absorption edge can be used to extrapolate the value of the direct bandgap of $HfS_2$.

**Photoresponse and scanning photocurrent microscopy measurements.** In our custom-made multi-functional opto-electronic setup, solid-state diode lasers are used and all the optical components are chosen in order to minimise deviations from the TEM$_{00}$ laser mode[25], which has a Gaussian intensity distribution. The lasers spot diameters ($d_s$) and depths of focus ($\Delta z$) are: for $\lambda = 375$ nm, $d_s = 264$ nm and $\Delta z = 158$ nm; for $\lambda = 473$ nm, $d_s = 445$ nm and $\Delta z = 268$ nm; for $\lambda = 514$ nm, $d_s = 484$ nm and $\Delta z = 291$ nm.

Scanning photocurrent maps were acquired by measuring the photo-generated current at each laser spot location ($\lambda = 473$ nm, $P = 150$ W cm$^{-2}$, see Supplementary Note 8 and Supplementary Fig. 8 for detailed electrical characterisation). The electrical signal from the device was amplified with a DL Model 1211 current preamplifier and measured with an Ametek Model 7270 DSP Lock-in amplifier. The locking frequency was provided by a function generator which modulated the lasers. The bias and gate voltages were provided by a Keithley 2400 SourceMeter.

**Band structure calculations.** First principles simulations were carried out using the density functional by Perdew, Burkeand Ernzerhof (PBE)[36], as implemented in the QUANTM ESPRESSO package[37]. The total energy of the system was

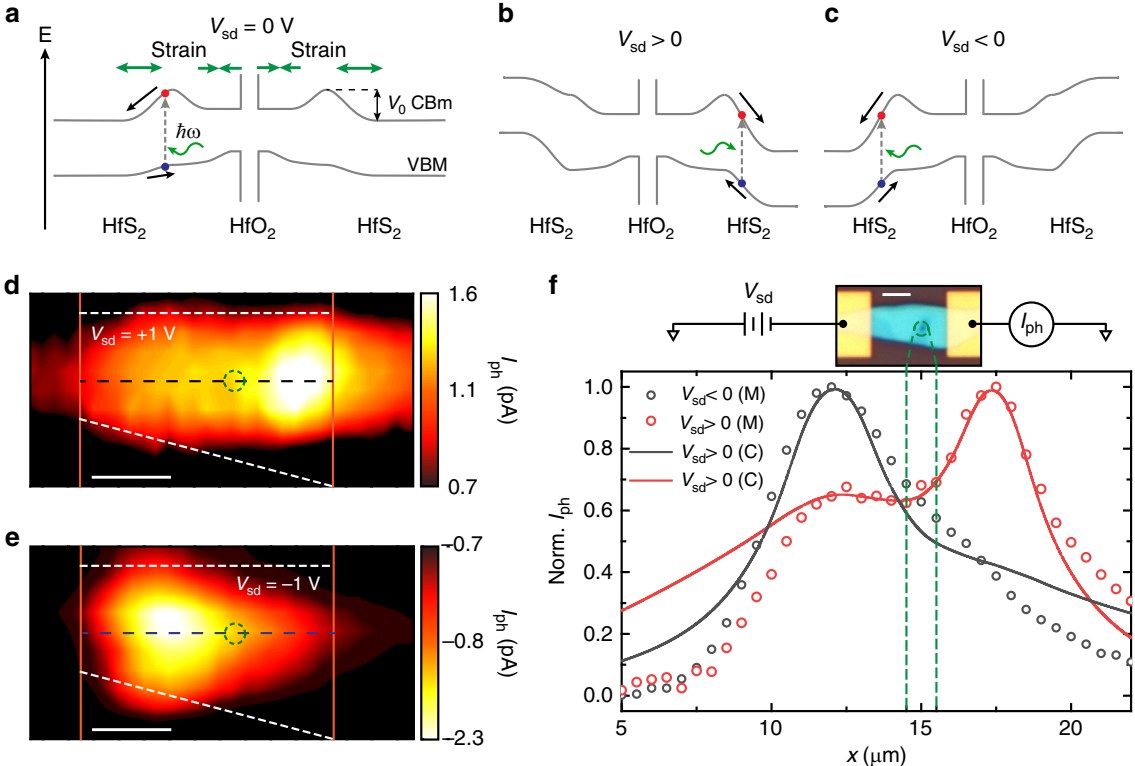

**Fig. 4** Charge funnel effect in $HfS_2/HfO_2$ engineered devices. **a** Schematic band diagram of a device subject to strain induced by local oxidation, with $V_{sd} = 0$ V, according to the proposed geometry in Fig. 1. **b, c** Schematic band diagrams of the device under $V_{sd} > 0$ and $V_{sd} < 0$, respectively. **d, e** SPCM map of the device under $V_{sd} = \pm 1$ V, respectively. SPCM maps were acquired using $\lambda = 473$ nm, $P = 150$ W cm$^{-2}$ at $V_{bg} = +30$ V. **f** Normalised photoresponse along the centre of the channel in the SPCM maps in panels **d, e** (dots) and simulated curves (solid lines) according to Eq. (1). Inset: optical micrograph of the measured device and measurement diagram. Scale bars are 3 μm

minimised with respect to coordinates of all atoms and the cell parameters for the bulk structures and the ground state obtained. For bulk, the structure was allowed to fully relax using the Broyden-Fletcher-Goldfarb-Shanno (BFGS) algorithm. A cutoff of 120 Ry and a $3 \times 3 \times 3$ Monkhorst-Pack k-point set were used for these calculations. Based upon these total energy calculations, reaction energetics were calculated (see Supplementary Note 7 and Supplementary Table 1). The reaction was taken as $HfS_2(s, 2D) + 3O_2(g) \rightarrow HfO_2(s) + 2SO_2(g)$, where the energy of the reaction is calculated from $E_R = E(HfS_2) + 3E(O_2) - E(HfO_2) - 2E(SO_2)$. The $HfO_2$ structure was fully relaxed.

**Data availability**. All data needed to evaluate the conclusions in the paper are present in the main text and/or the Supplementary Information. Additional data related to this paper are available from the corresponding authors upon reasonable request.

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

## Acknowledgements

The authors would like to thank R. Keens for support with the DFT calculations; M.D. Barnes and G.F. Jones for insightful discussion. S. Russo and M.F. Craciun acknowledge financial support from EPSRC (Grant no. EP/K017160/1, EP/K010050/1, EP/M001024/1, EP/M002438/1), from Royal Society international Exchanges Scheme 2016/R1, from European Commission (FP7-ICT-2013-613024-GRASP) and from The Leverhulme trust (grant title "Quantum Revolution" and "Quantum Drums"). I. Amit received funding from the People Programme (Marie Curie Actions) of the European Union's Eighth Framework Programme Horizon 2020 under REA grant agreement number 701704. S. Hepplestone acknowledges the use of the resources allocated by the MCC via membership of the UK's HEC Materials Chemistry Consortium, funded by EPSRC (EP/L000202), this work used the ARCHER UK National Supercomputing Service.

## Author contributions

A.D.S. conceived the idea, designed the experiment, performed the optical and electrical measurements. I.A. carried out the AFM measurements. S.P.H. carried out the DFT calculations. M.F.C. and S.R. supervised the project. All authors discussed the interpretation of the results and contributed to the writing of the paper.

## Additional information

**Competing interests:** The authors declare no competing interests.

