## [Peer Review File · Nature Communications]

REVIEWERS' COMMENTS:

Reviewer #2 (Remarks to the Author):

The response of the authors to the previous reports is convincing, in my view. I remain of the opinion that this work is worthy of publication in Nature and Nature Communications, and the corrections and extensions of the present submission strengthen the case even further. I reproduce my arguments in favor of publication here for completeness

The authors present the first demonstration of strain-engineered enhancement of the response in a phototransistor, fully consistent with the inverse funneling of photocarriers. The work has a powerful appeal in view of its implications in boosting the efficiency of solar energy harvesting, or in the field of single photon quantum emitters, which guarantees the interest of readers from quite a number of fields.

The experiment is furthermore a striking demonstration of a fundamental and unique property of 2D crystals: the high tunability of various electro-optical properties through strain engineering. In the field of 2D materials proper, I expect the results in this work to be received with great interest, as the concept of strain engineering underlies so much of the work being done on these materials and their appeal for applications in industry.

The work is well organised and clearly written. The motivation and conclusion are strong. The data is of very high quality. The experiment itself is carried out with great care.

[...]

My raised criticisms, particularly the discussion of theoretical efficiency limits in a solar cell context, have been convincingly addressed. I therefore recommend publications.

[A note: one of the assumptions in the second question of my report (that an unbiased case should necessarily produce a non-zero photoresponse in the presence of strain) was actually incorrect, as excitons or particle-hole pairs that are funneled as one (likely the case here) are charge neutral and therefore do not contribute to the photocurrent in this geometry unless a bias is applied. The experimental data confirms this too.]